# Tectonically-driven oxidant production in the hot biosphere

Jordan Stone ⬭[1], John O. Edgar ⬭[1], Jamie A. Gould[2] & Jon Telling ⬭[1] ✉

Genomic reconstructions of the common ancestor to all life have identified genes involved in $H_2O_2$ and $O_2$ cycling. Commonly dismissed as an artefact of lateral gene transfer after oxygenic photosynthesis evolved, an alternative is a geological source of $H_2O_2$ and $O_2$ on the early Earth. Here, we show that under oxygen-free conditions high concentrations of $H_2O_2$ can be released from defects on crushed silicate rocks when water is added and heated to temperatures close to boiling point, but little is released at temperatures <80 °C. This temperature window overlaps the growth ranges of evolutionary ancient heat-loving and oxygen-respiring Bacteria and Archaea near the root of the Universal Tree of Life. We propose that the thermal activation of mineral surface defects during geological fault movements and associated stresses in the Earth's crust was a source of oxidants that helped drive the (bio)geo-chemistry of hot fractures where life first evolved.

Studies tracing the common genes in Archaea and Bacteria to a Last Universal Common Ancestor (LUCA) have concluded that it was a thermophile or hyperthermophile ('liked it hot'), autotrophic (fixed $CO_2$), and dependent on $H_2$[1]. However, one contradictory feature of LUCA's inferred genome has been the presence of genes for cycling of $O_2$ and $H_2O_2$[1,2], despite models of UV-photochemical reactions in the Archaean atmosphere suggesting only trace (nM range) $H_2O_2$ would be present in surface waters on the early Earth[3]. The presence of these oxygen cycling genes has therefore been commonly explained as an artefact of the later evolution of photosynthetic oxygen and subsequent multiple lateral gene transfer events[1]. However, an alternative is that there was an additional more substantial geological source of $H_2O_2$ and $O_2$ in the Archaean prior to the evolution of oxygenic photosynthesis[4–6].

One potential geological source of $H_2O_2$ is the breakage of strong covalent bonds (≡Si-O-Si≡) during the crushing of silicate rocks (cataclasis), which produces an equal number of Si• and SiO• mineral surface free radical sites. The Si• are relatively labile and react with water to generate $H_2$ gas over hours to a week or more at 0 °C and above[7,8] (Eqs. (1) and (2); Supplementary Fig. 1).

$$\equiv Si\bullet + H_2O \rightarrow \equiv SiOH + H\bullet \tag{1}$$

$$2H\bullet \rightarrow H_2 \tag{2}$$

In contrast, SiO•, with a more oxidised $O^-$ rather than $O^{2-}$, have been shown to be relatively unreactive until much higher temperatures where they can prevent $H_2$ formation via reaction with precursor H• (Eq. (3))[7]. A recent study, however, has suggested that some reaction of SiO• on crushed quartz surfaces may be possible at room temperature[5]. Importantly, if SiO• can escape reaction with H• (Eq. 3) then they have the potential to react with water to generate $H_2O_2$[6,9] (Eqs. (4) and (5); Supplementary Fig. 1).

$$\equiv SiO\bullet + H\bullet \rightarrow \equiv SiOH \tag{3}$$

$$\equiv SiO\bullet + H_2O \rightarrow \equiv SiOH + \bullet OH \tag{4}$$

$$2\bullet OH \rightarrow H_2O_2 \tag{5}$$

A second geological source of $H_2O_2$ is from pre-existing intra-crystalline oxidised defects (peroxy bridges, Si–O–O–Si) within silicate rocks. These are formed during the cooling and crystallisation of magmas, where small quantities of water become incorporated into

[1]School of Natural and Environmental Sciences, Newcastle University, Newcastle-upon-Tyne NE1 7RU, UK. [2]Faculty of Science, Agriculture and Engineering, Newcastle University, Newcastle-upon-Tyne NE1 7RU, UK. ✉e-mail: jon.telling@newcastle.ac.uk

the crystal structures of igneous silicate minerals as hydroxyl groups[10]. The hydrogen of the hydroxyl group can be released as H• to form $H_2$ (Eq. (2)) which can diffuse out of minerals. In contrast, the remaining SiO• can pair with adjacent SiO• to form stable peroxy bridges. Additional peroxy bridges may be generated through time via α-recoil from α-radiation emitted by radionuclides such as U and Th concentrated within certain minerals[11]. On being uniaxially stressed (e.g. through tectonic forces in the crust) these peroxy bridges can break and migrate through mineral crystal structures, reforming SiO• at mineral surfaces[6] (Eq. (6)).

$$\equiv Si-O-O-Si\equiv \ \rightarrow\ 2\equiv SiO\bullet \qquad (6)$$

It has been proposed that peroxy linkages could have been a source of $H_2O_2$ to the subsurface prior to the evolution of photosynthesis[6]. It has been further suggested that mineral abrasion powered by streams, rivers and oceans would have generated SiO• and cleaved peroxy linkages on fractured silicate surfaces on the surface of the early Earth, perhaps generating sufficient $H_2O_2$ for ancestors of modern-day cyanobacteria to use $H_2O_2$ instead of $H_2O$ as a transitional electron donor to drive the evolution of oxygenic photosynthesis[5].

Here, we present experimental evidence demonstrating that temperature is an overlooked and important factor in maximising the yields of $H_2O_2$ from crushed silicate rock-water reactions under oxygen-limited conditions. We show that substantial $H_2O_2$ generation only takes place at elevated temperatures close to the boiling point of water, but importantly still within the documented thermal ranges of microbial growth (<122 °C[12]). We suggest that the combination of tectonic stress and heat generate sufficient hydrogen peroxide, and via disproportionation molecular oxygen, to potentially influence the ecology of microbial communities in the hot subsurface biosphere. We further suggest that this mechanism could have provided $H_2O_2$ on the early Earth, and influenced the chemistry of tectonically active hot subsurface fracture systems where life may have first evolved[1].

## Results and discussion
### Hydrogen and oxidant production
In an initial experiment, a single crushed rock (granite) was tested for its ability to generate $H_2$ over time (1 h, 1 day, 1 week) in a gastight $N_2$ flushed ball mill and temporarily heated to different temperatures (0, 30, 60 and 121 °C; Fig. 1; Supplementary Tables 2, 3, 4). The results for 1 h closely followed a previous study (Fig. 2) where granite was crushed for 20–30 min in a humid atmosphere, and where $H_2$ was interpreted to form via the reaction of mineral surface Si• defects with water (Eqs.

(1) and (2))[7]. A similar increase in $H_2$ from 40 °C to 100 °C has been shown in experiments with crushed pure silica after 30 min[13]. At higher temperatures after 30 min $H_2$ production increased up to 220 °C, then rapidly decreased[7] (Fig. 1). This is consistent with activation of SiO• at temperatures >220 °C resulting in the reaction of H• with SiO• (Eq. (3)), preventing $H_2$ formation.

Our data indicate that over timescales >1 h SiO• were reactive at 121 °C resulting in reduced $H_2$ generation (Fig. 1; Eq. (3)). This also suggests the further potential for SiO• to generate oxidants within the upper thermal limit of microbial growth (≤122 °C)[12]. To test this further, we carried out additional experiments crushing not only granite (a common rock in the continental crust since the Precambrian) but also basalt and peridotite (representing oceanic crust). The mineralogical compositions of the three rock types are given in Supplementary Fig. 2. As before, all crushing and manipulations were carried out under $N_2$, but using a more focused range of continuous incubation temperatures (60 °C, 80 °C, 104 °C, and 121 °C), and measuring oxidants ($H_2O_2$ and •OH) in addition to $H_2$.

Crushing generated similar concentrations of Si• across all three rock types (ranging from 13.0 to 14.4 μmol g⁻¹; Supplementary Fig. 3); sufficient to generate the maximum measured $H_2$ (c. 3 μmol g⁻¹; Fig. 2; Supplementary Table 3) via Eqs. (1) and (2). All three rocks were crushed to a similar final mean grain size (20.6–22.8 μm; Supplementary Fig. 4), with estimates of contamination from the agate ball mill and agate grinding balls to the rock powders ≤0.2% (Supplementary Table 5). We do not exclude the possibility that some of the $H_2$ in these (Fig. 2) and initial experiments (Fig. 1; Supplementary Fig. 5) was derived from other mineral–water reactions, in particular serpentinization-type reactions of $Fe^{2+}$ with $H_2O$[14]. We note however that the maximum aqueous $Fe^{2+}$ concentrations were three orders of magnitude lower than the maximum $H_2$ concentration, and that there was no significant ($p > 0.05$) correlation between $Fe^{2+}$ and $H_2$ ($R^2$ values ranging from 0.14 to 0.17; Supplementary Fig. 6). After a week, all three rocks produced significant $H_2$ above blanks at 60 °C and granite and peridotite produced significant $H_2$ at 80 °C (Mann-Whitney U: $P < 0.05$; Fig. 2). All rocks produced insignificant $H_2$ at 104 °C (Mann-Whitney $U$: $P < 0.05$; Fig. 3) suggesting that SiO• were primarily reacting between 80 and 104 °C. The significant decrease in $H_2$ production at 104 °C (ANOVA: $F_{2,24} = 6.408$, $P = 0.006$; LSD: $P = 0.005$ and $P = 0.005$) coincided with significantly higher $H_2O_2$ production (ANOVA: $F_{2,24} = 6.475$, $P = 0.006$) compared to 60 °C (LSD: $P = 0.004$) and 80 °C (LSD: $P = 0.005$). Wavelength-dependent absorption spectra of the coloured complex used to detect $H_2O_2$ in our analyses were a near identical match to those measured in $H_2O_2$ standards of a comparable

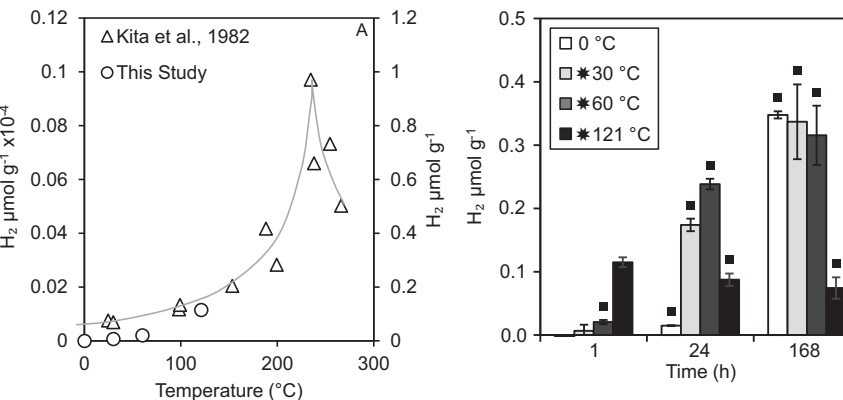

**Fig. 1 | $H_2$ production from crushed granite flash heated to 30 °C, 60 °C, or 121 °C compared to Kita et al.[7].  A** $H_2$ production from crushed granite measured by ref. 7. (left axis) compared with this study (right axis). The data from ref. 7 were measured at 20–30 min, while our study measured the data 1 h after heating (incubation at 0 °C). The line represents a trendline for data of ref. 7. and data are accurate to 15%. Error bars for this study are the standard error of the mean (not visible). **B** $H_2$ production 1, 24 and 168 h after flash heating of granite to 30, 60 or 121 °C (0 °C incubation). Concentrations are blank corrected, and bars with squares are significantly different from the blanks (Mann-Whitney U: $P < 0.05$). The error bars are the standard error of the mean.

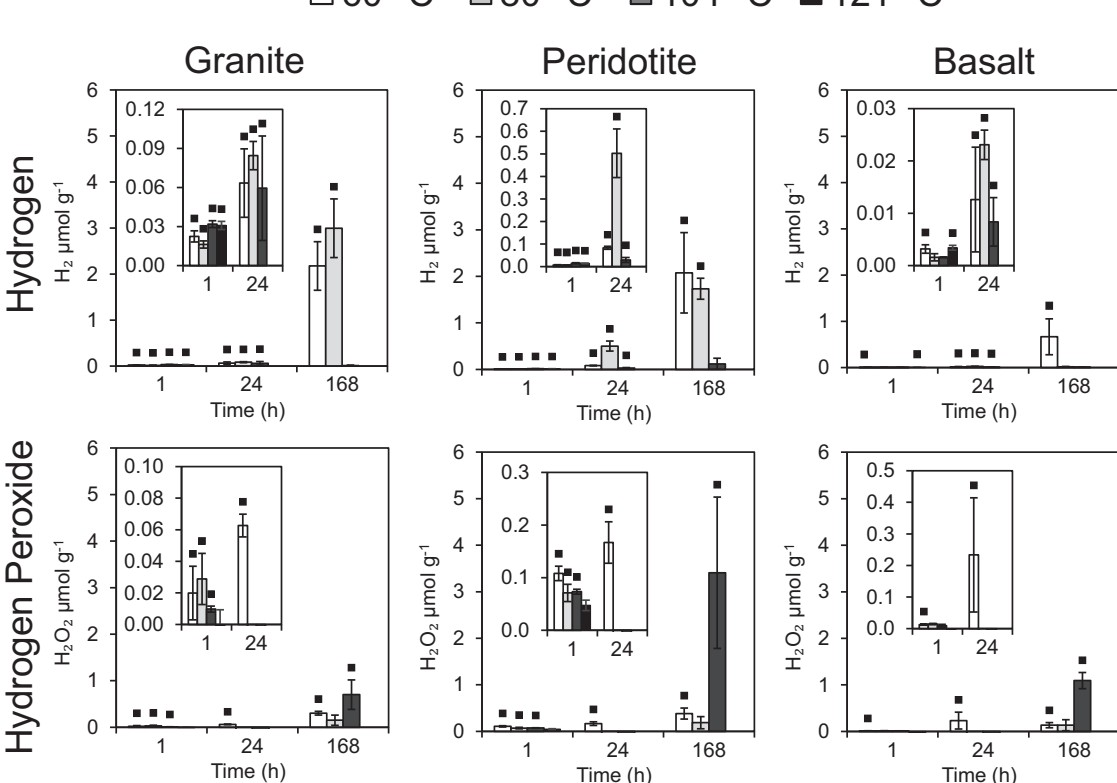

Fig. 2 | H$_2$O$_2$ and H$_2$ production from reactions of water with crushed granite, peridotite and basalt. All data are blank subtracted. A black square indicates a significant difference from the blanks (Mann-Whitney U: $P < 0.05$). Error bars are the standard error of the mean. The 121 °C experiments were measured after a mean of 2.9 h with a standard deviation of 0.2 h and at no other time point (no 121 °C data for 24 or 168 h). All other time points are accurate to the nearest whole number. The detection limits for H$_2$ and H$_2$O$_2$ were 0.2 nmol g$^{-1}$ and 9.2 nmol g$^{-1}$, respectively.

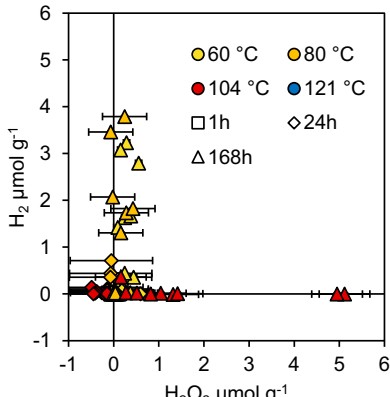

Fig. 3 | H$_2$O$_2$ and H$_2$ are produced from reactions of water with crushed rocks. The data are blank subtracted. The detection limits for H$_2$ and H$_2$O$_2$ were 0.2 nmol g$^{-1}$ and 9.2 nmol g$^{-1}$, respectively. The 121 °C experiment was only measured after 1 h, while the other temperatures show three time points: 1, 24 and 168 h. The error bars are 2× the standard deviation of the relevant blanks at each temperature and time point. Yellow symbols represent 60 °C, orange symbols represent 80 °C, red symbols represent 104 °C and blue symbols represent 121 °C. Squares represent 1 h, diamonds represent 24 h and triangles represent 168 h.

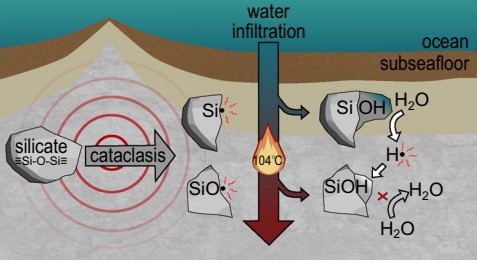

Fig. 4 | In subsurface environments at stable high temperatures (e.g. 104 °C), neither significant H$_2$O$_2$ or H$_2$ are produced. H• is produced from reactions of water with Si•, but it is removed by reactions with SiO• (Eq. (3)). These reactions result in the depletion of SiO• as SiOH forms, inhibiting H$_2$O$_2$ production from SiO•.

concentration (Supplementary Fig. 7), and to prior published wavelength scans from the original method[15]. This gives confidence that our H$_2$O$_2$ analyses were not an artefact of another compound interfering at the same wavelength of analysis. In contrast, no detectable •OH was measured (T-test: $t_{32.233} = 1.46$, $P = 0.154$), consistent with its role as a highly reactive intermediary. There was a clear trend showing the inhibition of H$_2$ production at 104 °C, coinciding with the enhanced production of H$_2$O$_2$ (Figs. 3, 4; Supplementary Fig. 8). Granite, peridotite, and basalt generated means of 0.70, 3.44 and 1.13 µmol g$^{-1}$ H$_2$O$_2$, respectively, after 1 week at 104 °C (equivalent to 171, 836, and 299 µM). The presence of detectable concentrations of H$_2$ and H$_2$O$_2$ in blank vials (water without crushed rock) (Supplementary Fig. 9) is consistent with the presence of preexisting Si•, SiO• and SiOO• defects within the borosilicate glass, as previously demonstrated in electron paramagnetic resonance studies of amorphous silica[16–18]. Additional surface Si• and SiO• defects may have been generated during the preparatory furnacing of our experimental vials (Methods) during the

dehydroxylation of the glass surfaces[19]. In the presence of oxygen in air, Si• can be readily converted at room temperature to reactive superoxides (SiOO•)[7,20] (Eqs. (7)–(9)) which can then react with water to generate $H_2O_2$[7,20].

$$\equiv Si\bullet + O_2 \rightarrow \equiv Si-O-O\bullet \tag{7}$$

$$\equiv Si-O-O\bullet + H_2O \rightarrow \equiv Si-OH + HO_2 \tag{8}$$

$$HO_2 \rightarrow 1/2\,H_2O_2 + 1/2\,O_2 \tag{9}$$

The reaction of Si• (which could otherwise have generated $H_2$ via Eqs. (1) and (2)) with oxygen to generate reactive superoxides and then with water to generate $H_2O_2$ therefore provides explanations for why $H_2O_2$ concentrations in blanks greatly exceeded that of $H_2$, and why $H_2O_2$ generation in blanks demonstrated a lesser dependence on higher temperatures relative to crushed rock (Supplementary Fig. 9).

Mass balance calculations (Supplementary Discussion) demonstrate that trace oxygen in crushed rock experiments, after correcting for blanks, could not account for more than 0.073 µmol g$^{-1}$ $H_2O_2$. One source of $H_2O_2$ could be the reaction of SiO• with water (Eqs. (4) and (5)). Previous work has suggested a mineral structure control on $H_2O_2$ generation when silicates are crushed under air, with an inverse relationship between the number of shared corners between silica tetrahedra and $H_2O_2$ generated[21]. The mean concentrations of $H_2O_2$ generated from the three rock types (peridotite > basalt > granite) in our experiments are broadly consistent with this order, since the dominant minerals in granite such as quartz and feldspars (tectosilicates) have greater numbers of tetrahedra sharing corners than the pyroxenes, amphiboles (inosilicates) and olivine (nesosilicates) present within the basalt and peridotite samples (Supplementary Fig. 2). However, during the cleavage of silicate bonds an equal number of Si• to SiO• are generated. At 104 °C, SiO• might therefore be expected to quantitatively react with an equivalent number of moles of H• generated from Si• (Eq. (3)) negating the potential for $H_2O_2$ generation[7]. An exception to this would be if there had been a significant prior reaction of Si• with water (either adsorbed on surfaces or released from mineral crystal structures[7]; Supplementary Fig. 10) to form $H_2$ during grinding in the ball mill, leading to an excess of SiO• over Si•. However, $H_2$ generated within the ball mill (Supplementary Fig. 11) accounted for only 0.4–3.4% of the excess oxidant generated after 1 week at 104 °C (Supplementary Figs. 12, 13).

A likely source of additional oxidants is peroxy bridges (Si–O–O–Si); ubiquitous oxidised defects within crystalline igneous and metamorphic rocks[5,6,10,22]. The average concentration of peroxy bridges within crystalline igneous and metamorphic rocks is 100 ppm[6]; equivalent to 2940 nmol g$^{-1}$. Since 1 mol of peroxy bridges can generate 1 mol of $H_2O_2$ on reaction with water (Eqs. (4), (5) and (6)), the activation and reaction of peroxy bridges are the correct magnitude to explain the concentration of net $H_2O_2$ in our experiments (means of 667 to 3273 nmol g$^{-1}$ at 104 °C; Fig. 2).

Our data are therefore consistent with the reaction of SiO•, generated both from cataclasis and peroxy bridges, to consume H•[7] and ultimately produce $H_2O_2$ at 104 °C on a timescale of ≤1 week. In addition, any OH• and $H_2O_2$ generated from SiO• sites that escaped reaction with H• may have reacted with $H_2$ via the 'Allen chain reaction' (Eqs. (10)–(12))[23,24].

$$H_2 + HO\bullet \rightarrow H\bullet + H_2O \tag{10}$$

$$H\bullet + H_2O_2 \rightarrow \bullet OH + H_2O \tag{11}$$

$$H_2 + H_2O_2 \rightarrow 2H_2O \tag{12}$$

While there are several reports suggesting the room temperature generation of $H_2O_2$ from the reaction of SiO• with water, the majority of these have either carried out the crushing of silicate minerals in air, and/or addition of water in air[6,21,25]. As noted above, however, $H_2O_2$ generation from crushed silicate minerals is greatly increased in the presence of $O_2$[5,20] due to the reaction of mineral surface defects with $O_2$ to generate the more reactive superoxide radical (SiOO•; Eq. 7) which can then react with $H_2O$ at room temperature to form $H_2O_2$ (Eqs. (8) and (9))[19]. Notably, further generation of $H_2O_2$ via this $O_2$-mediated pathway has been shown to cease after minerals are heated in water for 24 h at 60 °C[20]. In contrast, the continued production of $H_2O_2$ in our 104 °C experiments after 24 h (Figs. 2, 3) is consistent with a source from more stable SiO• rather than SiOO•[7,20].

Two room temperature studies that have crushed both quartz and added water under a nitrogen atmosphere have generated either no detectable $H_2O_2$[20], or much lower $H_2O_2$ concentrations (30 and 36 nmol g$^{-1}$ at sediment loadings of 0.4 and 0.6 g mL$^{-1}$ [5]), compared to 100 s to 1000 s nmol g$^{-1}$ at 104 °C at a sediment loading of 0.5 g mL$^{-1}$ in our experiments (Figs. 2, 3). The differences in the energy of crushing may have contributed to the differences in $H_2O_2$ generation in the two room temperature studies. $H_2O_2$ was detected in experiments when quartz was crushed for 5 h in a planetary ball mill at 350 rpm[5], generating high mineral surface areas, but not after lower energy 'end-over-end' abrasion over a longer period of time[20]. In addition, high-energy planetary ball milling[5] may increase the temperature of mineral surfaces during the crushing process. After our planetary ball milling (30 min at 500 rpm) the bulk temperatures of the crushed rocks increased by 10.7 ± 0.6 °C (Methods). If sufficiently high local temperatures were reached on mineral grain contacts during high energy milling, some SiO• sites may have been activated in an analogous manner to our higher temperature experiments (Fig. 3), with the water for reaction instead derived from either trace water on the ball mill or from within the structure of the crushed minerals[7]. Any $H_2O_2$ generated from SiO• within the ball mill that avoided prior reaction with H• (Eq. (3)) might then be expected to be released in a rapid pulse (<1 min) on the addition of the crushed mineral to water. We cannot discount that this may have contributed to the low concentrations of $H_2O_2$ detected after 1 h in our experiments, although these concentrations were negligible relative to the concentrations generated in our subsequent incubation experiments at elevated temperatures and longer time scales (Fig. 3). The potential for such milling-induced artefacts should, however, be considered in future room temperature crushed mineral-water experiments.

Importantly, the $H_2O_2$ reported in the prior room temperature study[5] was also associated with the generation of trace $O_2$, and around half of the $H_2O_2$ was inferred to derive from the mechanical cleavage of peroxy bridges (Eq. 13) and subsequent room temperature reaction of SiOO• with water[7] (Eqs. (8) and (9)).

$$\equiv Si-O-O-Si\equiv \rightarrow \equiv Si-O-O\bullet + \bullet Si \equiv \tag{13}$$

A component of the $H_2O_2$ measured in our experiments may therefore also have been generated from the heterologous cleavage of peroxy bridges (Eq. (13)). However, as noted above the dominant release of $H_2O_2$ at 104 °C indicates that more stable SiO•[7] rather than SiOO• was the primary source of $H_2O_2$ (Figs. 2, 3).

## Implications for life in tectonically active regions

The reaction of SiO• with water at ≤104 °C on a timescale of ≤1 week has important implications for the biogeochemistry of hot subsurface microbial ecosystems. First, it provides a previously unrecognised temperature barrier for mechanochemical $H_2$ generation. This is

consistent with measured rates of microbial activity in, for example, the seismically active subduction zone of the Nankai trough, where microbial methanogenesis ceases at 85 °C[26]. It also explains previous enigmatic experimental data which measured $H_2$ concentrations after crushed basalt, inoculated with a microbial slurry, was incubated from 2 °C to 110 °C for 130 days[27]. In this prior study, $H_2$ generation increased with temperature up to ~90 °C, followed by negligible $H_2$ production in the range of 105 to 110 °C after 130 days[27] (see Fig. 1 in ref. 27). There was also negligible $H_2$ production in autoclaved (121 °C) controls[27]. These results were interpreted as evidence of microbial catalysis of $H_2$ production, although the mechanism was unclear[27]. We reinterpret these results as consistent with first-order controls of abiotic $H_2$ production up to c. 90 °C due to the reaction of Si• with water (Eqs. (1) and (2)), followed by activation of SiO• at higher temperatures consuming $H_2$ (Eq. (3)). This interpretation is consistent with our experimental data i.e. the activation of SiO• at temperatures >80 °C, and extends these conclusions to a longer timescale (several months versus 1 week in our experiments).

The second important implication is that substantial concentrations of $H_2O_2$ can be generated from fractured silicate mineral-water reactions at 104 °C without the requirement for atmospheric $O_2$, but relatively little is generated at temperatures ≤80 °C (Fig. 3). We note that our measured $H_2O_2$ concentrations may underestimate total production due to subsequent decomposition[25], particularly if the decomposition is accelerated by higher temperatures and/or the presence of Fe species via Fenton reactions[28]. The presence of low concentrations of detectable dissolved iron species (up to nearly 0.037 μmol g⁻¹ (18.5 μM) in peridotite experiments) suggests the potential for Fenton reactions to have accelerated the removal of $H_2O_2$ (Eqs. (14) and (15))[28], although there were no clear trends with time or temperature in the Fe data (Supplementary Fig. 14), nor any significant relationship between $Fe^{2+}$ and $H_2O_2$ concentrations (Supplementary Fig. 15).

$$Fe^{2+} + H_2O_2 \rightarrow Fe^{3+} + HO\bullet + OH^- \tag{14}$$

$$Fe^{3+} + H_2O_2 \rightarrow Fe^{2+} + HOO\bullet + H^+ \tag{15}$$

Here, we make some global estimates of the potential importance of thermal SiO• activation in generating oxidants. We first make an estimate of the potential modern day subsurface flux of oxidants derived from the reaction of SiO• generated via cataclasis (Eqs. 3 and 4) by using prior estimates of global $H_2$ generation from the reaction of Si• (Eqs. 1 and 2) and assuming an equal molar ratio of SiO• generation; since there should be an equal number of Si• and SiO• sites generated during silicate fracturing. While global fluxes of $H_2$ generation from cataclasis are poorly constrained, a conservative estimate of $3 \times 10^{11}$ mol $H_2$ a⁻¹ has been calculated based on models of $H_2$ production as a function of global earthquake frequency and magnitude ($M$) (using $M$ values from 0 to 10[29] with $H_2$ yields based on experimental data[30]). This is a conservative estimate as significant $H_2$ generation may also be associated with substantial seismic activity along faults[29,31]. To showcase the potential maximum energy that this flux of oxidants could help release in the subsurface, we then assume that all of the oxidants potentially generated from SiO• during cataclasis are in the form of $H_2O_2$ (Eq. (16)) and multiply the $\Delta G_R^{100\,°C}$ of Eq. (16) (calculated using thermodynamic data from ref. 32) by the flux of $H_2O_2$ ($3 \times 10^{11}$ mol a⁻¹) to give $1.1 \times 10^{14}$ kJ a⁻¹, or 3.4 gigawatts (GW).

$$2H_2O \rightarrow H_2 + H_2O_2 \quad \triangle G_R^{100\,°C} = +353.03\,kJ\,mol\,H_2^{-1} \tag{16}$$

This is equivalent to 9.1% of organic matter-driven respiration in present day global marine sediments (37.3 GW)[33]. By normalising the estimated flux of $H_2O_2$ ($3 \times 10^{11}$ mol $H_2O_2$ a⁻¹) to the Earth's surface area ($5.011 \times 10^{18}$ cm²) and multiplying by Avogadro's constant ($6.022 \times 10^{23}$ mol⁻¹), this is equivalent to a potential flux of $1.1 \times 10^9$ molecules $H_2O_2$ cm⁻² s⁻¹. Importantly, however, rather than generating $H_2O_2$ our experiments and previous data[27] instead suggests that the SiO• oxidants generated via cataclasis at temperatures >90 °C (Fig. 2) will instead dominantly be used to negate mechanochemical $H_2$ production (Eq. (3); Fig. 4).

From the perspective of subsurface microorganisms, the bulk of this potential oxidant supply from cataclasis may therefore go to waste. We propose, however, that important exceptions may occur in environments where temperatures start cooler, allowing generation of $H_2$ from Si• defect sites, then increase through the temperature stability point of SiO• (Fig. 5). This could occur either through the burial and subsequent geothermal heating of tectonically crushed rocks or sediments, or due to temporally and spatially fluctuating temperatures in tectonically active regions, such as those associated with transform faults along mid-ocean ridges such as the Lost City Hydrothermal field where fluid flow paths can constantly evolve[34,35] (Fig. 5).

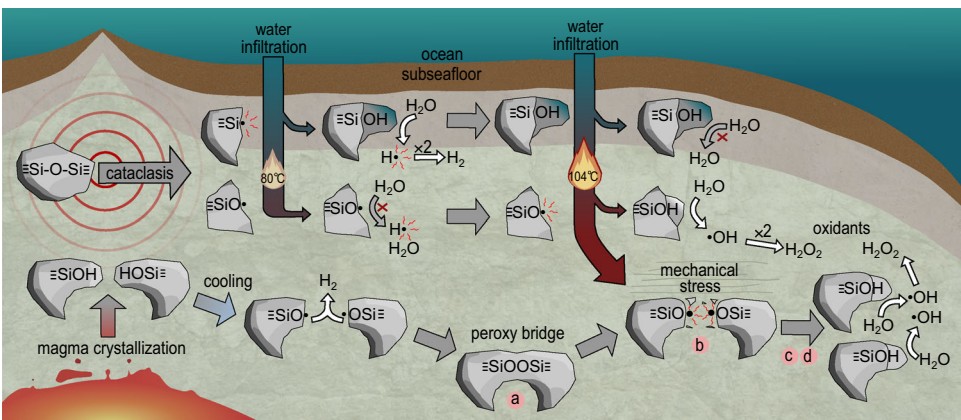

**Fig. 5 | Two anoxic mechanochemistry mechanisms leading to hydrogen and oxidant production in subseafloor environments.** The first mechanism is caused by earthquake-derived cataclasis, which forms Si• and SiO• from crushed ≡Si-O-Si≡. Hydrogen is formed at ≤80 °C from Si• and $H_2O_2$ is formed from SiO• at 104 °C. This mechanism requires fluctuating temperatures to produce $H_2O_2$. The second mechanism results from the formation of SiOH from magma crystallisation, releasing H• (which combine to form $H_2$) when cooled. A peroxy bridge subsequently forms, splitting into two SiO• under mechanical stress. Under stress, an electron from an adjacent $O^{2-}$ site can be transferred to the electron-deficient SiO•, which can be repeated in a chain reaction through the mineral[22]. This allows the migration of $O^-$ defects through the crystal and adjacent crystal boundaries to form surface SiO•[6,22], which can then react with water to generate $H_2O_2$ at 104 °C.

As an illustration of the potential importance of peroxy bridge SiO• to produce oxidants at a global scale in the present day, we consider their role during the tectonic cycle from crustal formation at mid-ocean ridges, to eventual destruction at subduction zones (Fig. 5). We assume a typical starting concentration of 100 ppm peroxy bridges[6] (equivalent to 2941 nmol g$^{-1}$ $H_2O_2$ production capacity), an annual production/subduction rate of oceanic crust of 19 km$^3$ [36], and an average density of oceanic crust of 3 g cm$^{-3}$. If we assume that all peroxy bridges are ultimately released by the combination of stress[6] and temperature (Fig. 2) to generate $H_2O_2$, this would generate $1.67 \times 10^{11}$ mol $H_2O_2$ a$^{-1}$, equivalent to $6.4 \times 10^8$ molecules $H_2O_2$ s$^{-1}$ cm$^{-2}$ when normalised over the surface of the Earth. Fluxes of $H_2O_2$ from both cataclasis and peroxy bridges, therefore, have potential to impact the subsurface biogeochemistry around active fault zones in the present day. Importantly, $H_2O_2$ generated from either cataclasis or peroxy bridges (Fig. 5) will be focused into localised fractures where by analogy with modelled concentrations of cataclastic $H_2$ generation[30] 100 s µM to mM concentrations of $H_2O_2$ could be locally generated; consistent with the results of our 104 °C experiments (Figs. 2, 3).

To put the modern-day estimates of $H_2O_2$ generation in context of the early Earth, the estimated maximum present day surface normalised fluxes from cataclasis ($1.1 \times 10^9$ molecules $H_2O_2$ cm$^{-2}$ s$^{-1}$) and peroxy bridges ($6.4 \times 10^8$ molecules $H_2O_2$ s$^{-1}$ cm$^{-2}$) are 1000× and 640× greater than that estimated to have been produced in the pre-photosynthetic Archaean atmosphere by UV photochemical reactions ($10^6$ molecules $H_2O_2$ cm$^{-2}$ s$^{-1}$)[3]. Clearly, however, the tectonic regime of the early Earth would have been very different to that of the present day. The date at which global plate tectonics similar to that of the present day first started is still an area of controversial active research, with estimates between 3.8 and 0.7 Ga depending on different lines of evidence[37]. Early recycling rates of plates may also have been appreciably faster than in modern tectonics[37]. Before global plate tectonics commenced, localised rates of subsurface stressing and fracturing, and hence potential $H_2O_2$ generation, would still have been present via more localised vertical tectonics[38], magma intrusion[39], and meteorite strikes[21]. The composition of Hadean and early Archaean crust would also have been very different and included rock types such as ultramafic komatiites that are not formed in the present day[40]. Our experimental results, with $H_2O_2$ generated from modern rock types spanning felsic to ultramafic (Fig. 2), suggest that oxidant production from cataclasis and peroxy bridges should also be relevant to these other silicate rock types.

Prior to the advent of oxygenic photosynthesis, we, therefore, propose that SiO• may have been an important source of $H_2O_2$ to hyperthermophilic microbial ecosystems in tectonically active regions. The presence of genes for cycling $H_2O_2$ and $O_2$ in reconstructions of LUCA's genome[1,2] has been explained as an artefact of the later evolution of photosynthetic oxygen and subsequent multiple lateral gene transfer events[1]. We suggest instead that these genes were required by a hyperthermophilic LUCA to deal with, and potentially make energetic use of, the oxidants produced from the reaction of SiO• with water during the stressing and fracturing of the Earth's early crust. This is consistent with the deeply rooted phylogenetic branches of hydrogenotrophic microaerophiles of the order Aquificea (Bacteria) and *Pyrobaculum aerophilum* (Archaea)[41,42]. *Aquifex aeolicus* grows between 85 and 95 °C[41], and *P. aerophilum* from 75 to 104 °C[42] which coincides with the inferred instability of SiO• and the production of $H_2O_2$ in experiments (Figs. 2, 3). Both microorganisms can grow on just $CO_2$, $H_2$, and $O_2$ and possess enzymes for $O_2$ respiration in addition to catalase function to disproportionate $H_2O_2$ to water and $O_2$ (Eq. (17))[43].

$$H_2O_2 \rightarrow H_2O + 1/2\, O_2 \qquad (17)$$

The energy available from using $H_2O_2$ or $O_2$ as an electron acceptor to oxidise $H_2$ (353.03 kJ mol$^{-1}$ $H_2$ and −294.68 kJ mol$^{-1}$ $H_2$

respectively at 100 °C; calculated using data from ref. 32) is greater than e.g. using $CO_2$ as an electron acceptor in methanogenesis (−45.3 kJ mol$^{-1}$ $H_2$ at 100 °C); energy that could be used to drive growth and evolution in early life. The disproportionation of $H_2O_2$ to water and $O_2$ is itself exothermic (−101.08 kJ mol$^{-1}$ $H_2O_2$ at 100 °C), and the high heat capacity of $H_2O_2$ has been proposed as a driver for the initiation of regular thermal cycles on reaction with reduced sulphur compounds; a potential mechanism for thermal cyclic RNA denaturation and replication in a pre-cellular RNA world[44]. $H_2O_2$ has also been demonstrated to play a pivotal role as an oxidant in a protometabolic analogue of the citric acid or tricarboxylic acid cycle[45] and $O_2$ derived from $H_2O_2$ can accelerate the abiotic synthesis of amino acids[46].

Our results demonstrate that under oxygen-limited conditions 100 s to 1000 s nmol g$^{-1}$ of $H_2O_2$ can be released from defects on crushed igneous silicate rocks when water is added and heated to temperatures close to boiling point (104 °C), but little is released at temperatures <80 °C. The dominant source of the $H_2O_2$ is most likely from pre-existing oxidised peroxy bridge defects within the silicate minerals which migrate to the surface under strain during the crushing process[5,6,10,22], and can then react with water at 104 °C. Importantly, this temperature of $H_2O_2$ generation overlaps the growth ranges of some hyperthermophilic microorganisms, including evolutionary ancient heat-loving and oxygen-respiring microorganisms near the root of the Universal Tree of Life[42]. We suggest that the production of oxidants via the thermal activation of SiO• in tectonically active regions may influence the ecology of our present-day subsurface hot biosphere, and may also have influenced the biogeochemistry of subsurface fractures on the early Earth. While the date at which global plate tectonics started is an area of controversial active research[37], more localised rates of subsurface stressing and fracturing, and hence potential $H_2O_2$ generation, would still have been present on the early Earth via more localised vertical tectonics[38], magma intrusion[39], and meteorite strikes[21]. $H_2O_2$ has been proposed as a key molecule in some theories for the origin of life[44–46]. The thermal activation of oxidised mineral defects during geological fault movements and associated stresses in the Earth's crust may therefore have had a role as a source of oxidants that helped drive the (bio)geochemistry of hot fractures where life first evolved.

## Methods
### Initial flash heating experiment
Granite (Cumbria) was commercially sourced from Northern Geological Supplies Limited Cumbria, UK (catalogue number gasg1kg). It was initially shattered with a sledgehammer on an anvil (washed with 100% ethanol) within multiple thick polyethene bags. The rock was then crushed with a jaw crusher (after an initial discarded sample) and sieved to a consistent 1–3 mm size fraction. The 1–3 mm fraction was then washed in 18.2 MΩ cm$^{-1}$ water to remove fine dust particles until the residual water was visibly clear, and then dried at 60 °C for >1 week before milling. The rock fragments (45 g) were then crushed in a gastight stainless steel-encased agate ball mill within a Fritsch P6 Planetary Ball Mill[8]. The agate ball mill was cleaned twice by milling with pure quartz and once with the rock to be used in experiments for two minutes each at 500 rpm, before crushing the experimental rock sample. Prior to crushing, the agate mortar of the ball mill was sealed with a gastight agate lid containing a viton o-ring, enclosed within a custom-made stainless steel triaxial clamping system. The ball mill was then vacuumed and flushed with $N_2$ for seven cycles before equilibrating the $N_2$ headspace to atmospheric pressure using a gastight syringe[8]. Each milling was performed at 500 rpm (g-force: 34 g) for 30 min[8]. The ball mill was then transferred and opened within a glove bag filled and continually flushed with 5.0 grade $N_2$ (<10 ppm $O_2$). The $O_2$ within the glove bag was also confirmed to be <0.1% $O_2$ via a Presens optical $O_2$ sensor. 2 g (1.937−2.048 g) sub-fractions were then transferred into 10 mL borosilicate serum vials (previously autoclaved,

bathed in 10% HCl for 2 h, rinsed in 18.2 MΩ cm⁻¹ water, and furnaced at 500 °C for 4 h). Vials were sealed with thick butyl rubber stoppers (previously autoclaved at 121 °C for 30 min, boiled in 1 M NaOH for 1 h[8], rinsed in 18.2 MΩ cm⁻¹ water, and dried at 60 °C), and crimp sealed. The remaining rock powder was stored at room temperature in a sealed plastic tub for grain size analysis. The blank controls were treated identically but with the omission of the rock powder. The vials were then flushed with $N_2$ for two minutes each to remove any trace oxygen and then equilibrated to 1 atm [8].

De-oxygenated water was prepared by autoclaving 18.2 MΩ cm⁻¹ water in a 0.5 L borosilicate Duran bottle at 121 °C for 1 h. The Duran bottle was then placed in a 0 °C water bath, and vigorously bubbled with $N_2$ for ~4 h, and dissolved $O_2$ was measured to be 0.216 mg L⁻¹ (equivalent to 6.74 µmol L⁻¹) with a calibrated Presens $O_2$ optical sensor. 4 mL of de-oxygenated water was then added to each vial (~2 h after crushing) using a gastight syringe and needle and shaken for 10 s. The vials were stored in a 0 °C water bath for 5–30 min before heating. Batches of nine vials were either kept at 0 °C, or rapidly ('flash') heated to 30 °C (water bath) 60 °C (oven), or 121 °C (autoclave) for 1 min, then vials rapidly cooled in the dark at 0 °C. Triplicate vials were then destructively sampled at 1 h, 24 h and 168 h (Supplementary Fig. 16). During sampling, 4 mL gas (replaced with $N_2$) were taken and stored at an overpressure in 3 mL Exetainers with double wadded caps (evacuated to <0.6 mbar) using a gastight syringe and needle. The vial was then shaken to ensure that the liquid was homogenous, and a 2 mL liquid sample was extracted using a syringe and needle.

$H_2$ was measured using a ThemoFisher Gas Chromatograph with a He Pulsed Discharge Detector with a 2 m micro-packed Shin Carbon ST 100/120 mesh, 1/16 inch OD, 1.0 mm ID column, with a constant flow (10 mL⁻¹) of He carrier gas and a run-time of 12.5 min. The column temperature was 60 °C, the injector temperature was 110 °C, and the detector temperature was 110 °C. Samples were calibrated to certified (±5%) 100 ppm standards (BOC). Three standards were run daily throughout the experimental period (coefficient of variation = 5.97%, $n = 39$). 100 µL of gas from Exetainers was injected directly onto the column of the GC. The Ideal Gas Law was used to calculate the mol $H_2$ in the headspace of the vials from the ppm concentration[9]. Adjustments were made for dilutions during sampling, and the mol $H_2$ normalised to dry material weight.

The grain size was measured by laser diffraction using a Mastersizer 3000 using the software Mastersizer v3.81, with 15 replicates per analysis. Three separate samples from the material that remained after milling were added to the Mastersizer at 10–20% obscuration with five replicates each.

## Continuous heating experiments

In addition to granite (Cumbria, UK, catalogue number gasg1kg), basalt (Isle of Skye, UK, catalogue number bas1kg) and peridotite (Finland, catalogue number IGNROK043) were crushed in a further experiment to measure oxidants in addition to $H_2$ under a more targeted range of continuous heating experiments. The additional rocks (basalt and peridotite) were chosen as they are dominant constituents of oceanic crust. Granite, in contrast, has been a common rock of the continental crust since the Precambrian. All rock samples were commercially sourced from Northern Geological Supplies Limited. The same crushing process (including preparations and weights) was used for these experiments (i.e. 500 rpm for 30 min under an $N_2$ atmosphere).

In contrast to the pilot experiments, the final $N_2$ flushing step of the vials was not used since our $O_2$ monitoring confirmed only trace $O_2$ (<0.1% v/v) within the glove bag, and because of the potential for some material loss through the needle during the flushing step. Furthermore, the water was added at room temperature rather than at 0 °C, and the vials were heated to and incubated at either 60 °C, 80 °C, 104 °C or 121 °C for 1, 24 or 168 h, with the exception of 121 °C experiments which for logistical reasons were only heated for 1 h in an autoclave; Supplementary Fig. 17). The $O_2$ concentrations in the $N_2$-purged water added to experiments was ≤8.4 µmol L⁻¹.

$H_2O_2$ concentrations were analysed using a UV-spectrophotometric method[15]. The method measures the quantity of copper (I)–DMP complex (Cu(DMP)²⁺) that forms in the presence of $H_2O_2$, at a wavelength of 454 nm after ~20 min of reaction time. Daily standards were made from a stock solution of 1000 µM $H_2O_2$ to prevent the photodegradation of $H_2O_2$ (coefficient of variation = 0.03%, $n = 12$). The detection limit for $H_2O_2$ was 9.2 nmol g⁻¹. A 0.01 mol L⁻¹ solution of copper (II) sulphate was created by adding 0.2497 g of copper (II) sulphate to a volumetric flask, filled to 100 mL with deionised water. A solution of 1 g of 2,9-dimethyl-l,lO-phenanthroline (DMP) in 100 mL ethanol was also created (0.048 mol L⁻¹). 0.125 mL of each reagent was added to each 1 mL sample or standard in a cuvette and mixed. The samples were left for 20 min to allow the reaction to proceed. The concentration of copper (I)–DMP complex (Cu(DMP)²⁺) was quantified using a Biochrom Libra S12 UV–Vis spectrophotometer measuring at a wavelength of 454 nm.

$H_2$ in the continuous heating experiments was analysed on an SRI Gas Chromatograph with a dual mercury Reduction Gas Analyser (RGA) and Thermal Conductivity Detector (TCD) (in series). 2 mL of gas was injected onto a 0.5 mL sample loop which then injected the sample onto a 6 ft packed molecular sieve 5 Å column at a constant pressure of 20 psi $N_2$, column temperature of 40 °C, RGA temperature of 280 °C, and TCD temperature of 100 °C. Low concentrations (<1000 ppm) were quantified using peak areas from the RGD detector, higher concentrations (>1000 ppm) by the TCD detector. Samples were calibrated to certified (±2%) standards of concentrations 10, 100 or 20,000 ppm. The coefficient of variation of GC-SRI standards were 5.37%, 8.20% and 5.88% for 10, 100 and 20,000 ppm, respectively. The detection limit for $H_2$ was 0.2 nmol g⁻¹.

The concentration of •OH adsorbed to the rock surface was measured based on the reaction of •OH with pCBA[20]. Two 45 µM pCBA solutions were made, one in deionised water and the other in deionised water with 10% methanol (7.8 mg pCBA in 500 mL). Exetainers were evacuated, and then $N_2$ flushed for ~2 min. For each vial, an Exetainer with 1 mL of pCBA in water, and an Exetainer with 1 mL pCBA in water and methanol were prepared. Following removal of liquid for samples for $H_2O_2$ and Fe, the vial was shaken vigorously to ensure that all material was suspended in the liquid. Then a 0.5 mL sample of the slurry was removed using a 1 mL syringe and needle. 0.25 mL of slurry was added to each Exetainer. The Exetainers were left for 30 min to allow the reaction between pCBA and •OH to take place. The contents were filtered through 0.2 µm nylon filters and transferred to 1.5 mL glass autosampler vials. The concentration of pCBA was determined by a ThermoFinnigan Surveyor for high-pressure liquid chromatography (HPLC) with a quaternary solvent pump and photodiode array detector. The pump was running at 0.5 mL min⁻¹. The injection volume was 10 µL, injected on a sample loop-flex 100 µL loop at a temperature of 45 °C. The concentration of pCBA was determined using standards of pCBA dissolved in acetonitrile ranging from 0 to 50 µM (coefficient of variation = 9.09%). Peak integration was conducted with ChemStation and OpenChrom. The pH in experiments was also measured using Fisherbrand™ pH paper sticks (pH 0–14).

Thermogravimetric analysis (TGA) coupled with Differential Scanning Calorimetry (TG-DSC) was performed on a 40 mg aliquot of each crushed rock. The samples were heated from room temperature up to 1000 °C in an alumina crucible at a rate of 10 °C min⁻¹ in synthetic air (20% Oxygen/80% Helium) with a flow rate of 40 mL min⁻¹. The adaptor heater and transfer line were heated to 150 °C and 300 °C, respectively. The protective gas (Helium) flow was set at 25 mL min⁻¹ to enable stability of the weighing balance during analysis. The evolved gas during heating was measured in scan mode (10–150) by a Netzsch Quadstar 442 (QMS) coupled to the TG-DSC instrument in triggered run mode, with emphasis on $m/z$ 17 and 18. The differential scanning

calorimeter measured the difference in heat flow rate between the sample and an inert reference. The data was processed and exported to a Microsoft Excel file using Netzsch Proteus Analysis Software.

The ferrozine spectrophotometric method was used to measure $Fe^{2+}$ and $Fe^{3+}$ in the liquid samples[4,47,48]. The method measures the concentration of $Fe^{2+}$ based on the measurement of $Fe^{2+}$–ferrozine complex formed from the reaction of $Fe^{2+}$ with ferrozine. The use of a reducing agent, hydroxylamine hydrochloride, allows the measurement of $Fe^{3+}$ concentration[47]. The samples were stored in pre-evacuated Exetainers for <2 weeks prior to analysis. Three reagents were created: A reducing agent, hydroxylamine hydrochloride (1.4 mol $L^{-1}$ in 2 mol $L^{-1}$ hydrochloric acid in deionised water), a buffer, ammonium acetate (10 mol $L^{-1}$ in deionised water and adjusted to a pH of 9.5 with an ammonium hydroxide solution), and a solution of 0.1 mol $L^{-1}$ ferrozine in DI water. Standards of $Fe^{3+}$ were created by diluting $FeCl_3$ to varying concentrations to a range of 0–23.81 μmol $L^{-1}$. 1 mL of sample and 100 μL of the ferrozine reagent were added to a cuvette. The cuvette was mixed, and the concentration of $Fe^{2+}$–ferrozine complex was measured at a wavelength of 562 nm in a UV–Vis spectrophotometer. Next, the $Fe^{3+}$ was reduced to $Fe^{2+}$ by the addition of the reducing agent and left for 10 min for the reaction to proceed. Finally, a buffer was added, and the concentration of $Fe^{2+}$–ferrozine complex was measured again.

## Measurement of $H_2$ generated during ball milling

To measure the $H_2$ generated during the grinding of the different rocks in the ball mill, we carried out separate runs in triplicate for each rock type using the same ball milling apparatus as for the main experiments. In each run 45 g of each rock (1–3 mm fraction) was ground in the gastight agate ball mill for 30 min at 500 rpm. The ball mill was transferred to the glove bag (<0.1 ppm $O_2$) and a 1/8 inch Swagelok fitting containing a rubber septa was attached to the one of the ball lid valves[49]. A 10 mL gastight syringe and needle was then used to sample 3 mL of gas from the ball mill, after pumping the syringe 6× to ensure the outlet valve gas was in equilibrium with the ball mill interior. This 3 mL of gas was injected into a 3 mL pre-evacuated Exetainer with double-wadded cap, and $H_2$ analysed as described above for continuous heating experiments.

## Identification of mineral phases

Mineral phase identification of experimental materials was performed by XRD utilising a PANalytical X'Pert Pro MPD, powered by a Philips PW3040/60 X-ray generator fitted with an X'Celerator detector. Diffraction data were acquired by exposing powder samples to Cu-Kα X-ray radiation, which has a characteristic wavelength (λ) of 1.5418 Å. X-rays were generated from a Cu anode supplied with 40 kV and a current of 40 mA. Data sets were collected over a range of 5−100° 2θ with a step size of 0.0334° 2θ and nominal time per step of 1 s, using the scanning X'Celerator detector and a secondary Ni monochromator in the diffracted beam path. The optics set up for the instrument were as follows; programmable divergence slit with a fixed length of 10 mm, an incident anti-scatter slit of 4°, a beam mask of 20 mm and incident/diffracted Soller slits of 0.04 radians. All scans were carried out in 'continuous' mode. All XRD data were recorded as .XRDML files to which profiles were fit using a minimum 2nd derivative method and which were then evaluated by searching the Crystallography Open Database (COD; http://www.crystallography.net/cod/) using the Malvern Panalytical HighScore Plus software package. The search was restricted to incorporate phases containing at least one of the major elements O, Si, Al, Fe, Mg, Na, K and Ca and potentially containing Ti, Mn, Ni, S and Cl. The data were analysed further using Reitveld methods. The Rietveld method involves constructing a model consisting of the crystal structures of all component phases, and the differences between the observed and simulated diffraction patterns are minimised by varying scale factors, unit-cell parameters, and crystallite

size for each phase. This method provides information on well-ordered (crystalline) phases.

## Measurement of surface Si radicals

All three rocks were crushed again under the same conditions to measure the concentration of Si on the surface of the crushed rocks[9]. 5 mL of DPPH solution (50 mg DPPH, 1 L ethanol) was added to 30 mg of crushed rock under an $N_2$ atmosphere. After shaking, the solution was filtered through a 0.2 μm filter and left for 1 min. The concentration of DPPH lost was measured at 515 nm using a UV–vis spectrophotometer.

## Measurement of temperature change after ball milling

To assess bulk changes in temperature caused by the milling process, the three rock types were crushed once more and the temperature of the rocks and the ball mill casing noted before and after milling using a non-contact RS-820 infra-red thermometer.

## Estimation of agate contamination during ball milling

Further milling experiments were conducted to estimate the degree of contamination from the agate ball mill and agate grinding balls during the ball milling. The ball mill (including lid and grinding balls) was first cleaned with ethanol, dried with $N_2$ and weighed on a Mettler PE11 weighing scale (precision ±0.1 g). 45 g of each of the three rock types (granite, peridotite and basalt) were then ground at 500 rpm for 30 min. Between each run, the ball mill was emptied, re-cleaned with ethanol, dried with $N_2$ and re-weighed. The difference between the start and end weights represents the agate mass loss during milling.

## Data analysis

All concentrations were normalised to dry sediment mass (μmol $g^{-1}$). Schematic diagrams were produced using Inkscape. Plots and charts were generated using Microsoft Excel and exported using Inkscape. Detection limits were calculated based on the sum of the mean of the blanks and the standard deviation multiplied by three. Statistical analyses were conducted using IBM SPSS Statistics 25. Due to the presence of the 121 °C experiment in only one-time point in the first granite experiment, it was excluded from overall correlations and comparisons between temperatures and time points. To test whether each temperature and time point was significantly different from the blanks, Mann-Whitney U tests were used ($P < 0.05$; two-tailed). Independent samples T-tests ($P < 0.05$; two-tailed) were used to determine if, overall, the chemistry was significantly different from the blanks (e.g. for •OH). One-way ANOVA (two-tailed) was used to test for significant differences in $H_2$ and $H_2O_2$ production between temperatures after a week.

## Data availability

The data generated in this study have been deposited in the National Geoscience Data Centre available at: https://doi.org/10.5285/026721ce-4975-4628-8f69-807b78dd3fe4. Source data are provided with this paper.

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

## Acknowledgements

Thank you to Johnny Rutherford for assistance with sampling and analyses. We also acknowledge help from Alex Charlton with HPLC analyses, Lisa Deveaux-Robinson and Dave Earley for technical assistance in laboratories, Onos Esegbue for running TGA analysis, and Ana Contessa for aid in grain size analysis. This research was supported by UK Space Agency Aurora grants ST/R001421/1 and ST/S001484/1 (to J.T.), and NERC grants NE/S001670/1 and NE/W005506/1 (to J.T.).

## Author contributions

J.S. carried out the majority of laboratory work, helped design the detailed methods, conducted data analyses and initial interpretations, produced figures and co-wrote the manuscript. J.O.E. aided with the development of theories and concepts for the project and discussion, aided in chemical analyses, and contributed to the manuscript. J.A.G. conducted XRD analyses and aided with their interpretation. J.T. conceived the project idea and overall methodology, supervised lab work, conducted some of the laboratory work, and co-wrote the manuscript.

## Competing interests

The authors declare no competing interests.
