## [Peer Review File · Nature Communications]

Tectonically-driven oxidant production in the hot biosphereREVIEWER COMMENTS

Reviewer #1 (Remarks to the Author):

This is an interesting paper that reports the generation of H₂O₂ and H₂ on the crushed silicate rocks (i.e., granite, peridotite and basalt) when water is added and heated temperatures close to boiling point. This temperature window overlaps the growth ranges of evolutionary ancient heat-loving and oxygen-respiring Bacteria and Archaea near the root of the Universal Tree of Life. Thus, the authors propose that the thermal activation of mineral surface defects during geological fault movements and associated stresses in the Earth's crust was a source of oxidants that helped drive the (bio)geochemistry of hot fractures where life first evolved. However, the data analyses as well as the experimental results are not convincing, and there are several important issues that should be addressed clearly:

1. Surface radicals (e.g., $\equiv\text{Si}-\text{O}\bullet$, $\equiv\text{Si}\bullet$, and $\equiv\text{Si}-\text{O}-\text{O}\bullet$) of silicate minerals are derived from the homolysis of Si-O-Si bonds. One of the key points for this study is the formation of radicals during ball milling of the rocks. Thus, the authors should provide the evidence for the formation of the corresponding radicals (e.g., $\equiv\text{Si}\bullet$) in the crushed rocks.
2. In general, the formation of surface radicals in silicate minerals during ball milling strongly depends on Si polymerization degree (the ratio of bridging oxygen per silicon), which commonly decreases in the order of tectosilicate, phyllosilicate, cyclosilicate, inosilicate and nesosilicate. However, this study indicates that peridotite, mainly composed of nesosilicate, has a higher yield of H₂O₂ than granite in which quartz and feldspar (tectosilicate) are main component minerals. The experimental results seem to be unreasonable.
3. The chemical compositions of the rocks should be provided. Generally, both basalt and peridotite are rich in Fe²⁺ and Mg²⁺, belonging to mafic and ultramafic rocks, respectively. The reaction between basalt/peridotite and water can also produce H₂ (similar to serpentinization). Did the authors consider the contribution of this reaction to the total H₂? More importantly, Fe²⁺ readily reacts with H₂O₂ (i.e., Fenton Reaction), leading to consumption of H₂O₂ and oxidation of Fe²⁺. Thus, the contents of Fe²⁺ in the rocks and solutions have a prominent effect on the quantity of H₂O₂. These factors should be considered in the data analyses and the related discussions.

Reviewer #2 (Remarks to the Author):

In a manuscript entitled "Tectonically driven oxidant production in the hot biosphere", the authors present experiments that may shed new light on the role of O₂ and reactive oxygen species (ROS), mainly H₂O₂, in the earliest stages of the evolution of life on Earth and the formation of LUCA.

According to the prevailing view, the first LUCA protocells formed under strictly anaerobic conditions near hydrothermal vents (Weiss et al. 2016, *Nat Microbiol* 1:1). The absence of oxygen (O₂) and thus its reactive derivatives (ROS) in the environment did not require the presence of antioxidant enzymes to remove excess ROS in LUCA. On the other hand, phylogenomic data suggest that LUCA may have possessed genes encoding different O₂/ROS-utilizing enzymes (Jabłońska & Tawfik 2021, *Nat. Ecol. Evol.* 4:442). So the question arises: if LUCA was equipped with a primitive form of the enzymatic antioxidant system, what could have been the sources of O₂/ROS that generated the selection pressure that led to the evolution of antioxidant enzymes? On the one hand, there are possible abiotic sources of O₂/ROS: photolysis of H₂O and CO₂, radiolysis of H₂O or imports from space, for example via comets (Ślesak et al. 2019, *FRBM* 140:61). On the other hand, there are also phylogenomic data suggesting that the source of O₂ on the young Earth was also oxygenic photosynthesis, which may have occurred about 3.4 billion years ago (Oliver et al. 2021, *BBA Bioenerg.* 1862:148400). Either way, both the early occurrence of oxygenic photosynthesis and possible abiogenic sources of O₂/ROS are not mutually exclusive.

The authors hypothesise that an important abiotic source of H₂O₂ may have been cataclasis, the crushing of quartz rock at high temperatures. This process produces both H₂O₂ and H₂. In carefully designed and executed experiments, they showed that the production of H₂O₂ at 104 °C is relatively efficient. Already in the discussion of their results, the authors linked the production of H₂O₂ at high temperatures to conditions at hydrothermal vents and the fact that the primordial organisms may have been thermophilic/hyperthermophilic. Their findings would explain, at least in part, the abiotic sources of H₂O₂ on the early Earth and the possibility that O₂/ROS-utilizing enzymes evolved in the earliest life forms.

The main novelty of the work presented is that it reveals a potentially different source of O₂/ROS on Earth about 3.5 billion years ago, when life arose. This work complements recent studies showing that abrasion of quartz surfaces leads to the formation of O₂/ROS (He et al. 2021. *Nature* 12:1).

The methodological side of the work is good. Especially the use of blanks. This was particularly important because the so-called de-oxygenated water also contained O₂ that could interact with the rock material studied.

Minor comments for the authors' consideration.

1. p. 6, l. 164: The wording “[...] was warm to the touch [...]” seems somewhat inaccurate. It would be better to give at least an estimate of the temperature to which the ball mill heated.
2. Discussion on p. 9: The H₂O₂ produced by cataclasis could also be involved in the prebiotic synthesis of e.g. amino acids. Experiments on the synthesis of amino acids under hydrothermal conditions have also shown that the addition of O₂/H₂O₂ can also increase the rates of abiotic synthesis of selected amino acids and allow the synthesis of others, especially amino acids with additional hydroxyl groups, such as Ser, Lys and Thr (Marshall 1994, *Geochem. Cosmochim. Acta* 58: 2099). Therefore, a role of local

microquantities of O₂/H₂O₂ in the processes leading to the prebiotic synthesis of important biopolymers cannot also be completely excluded.

3. A minor methodological note concerns the representation of rock crushing in a ball mill in revolutions per minute ("rpm") and not in "g" (g-force, the relative centrifugal force [RCF]). In centrifuges, "g" is a more universal unit, as "rpm" depends on the radius of the rotor used. I am not sure, but perhaps for ball mills only "rpm" is used as a unit.

4. Add a legend to Supplementary Figure 2.

Concluding remark

In my opinion, this manuscript should be accepted for publication in Nature Communications.

Ireneusz Ślesak

Reviewer #3 (Remarks to the Author):

This study challenges the common notion that the Early Earth's oceans contained essentially no oxygen and hydrogen peroxide until it was produced as a microbial waste product when life was already firmly established. By contrast, the genome of the Last Universal Common Ancestor (LUCA) does contain genes that encode for metabolizing O₂ and H₂O₂. Without the presence of these oxidants at the time of the emergence of life, it is difficult to understand why this capability was encoded for. Hence, along with the common notion that O₂ and H₂O₂ were essentially absent it is assumed that the presence of this metabolic capability in LUCA is the result of lateral gene transfer, reflecting the widespread acquisition of this capability across life well after the emergence of life.

As referenced in this study, atmospheric production of H₂O₂ would have produced a low flux of H₂O₂ into the ocean where life likely emerged. The present experimental study indicates that there may be two alternative pathways. One hinges on breaking bonds within common silicate rocks that lead to the formation of surface radicals. The other is based on the formation of peroxy bonds formed during the crystallization of melts. Both mechanisms have been studied in detailed, but this study has focused on the magnitude of the formation of H₂ and H₂O₂. The techniques and interpretation of the results are well documented. A key innovative aspect is the grinding of the materials under exceedingly well-controlled conditions that exclude water vapor and oxygen. The selection of rocks is appropriate, although the early earth probably did have little granite and was likely dominated by komatiite (Taylor SR, McLennan SM (2009) Planetary Crusts: Their Composition, Origin and Evolution. Cambridge University Press, 378 pp). However, the exact composition of the rock type is likely not a major factor in the outcome as it hinges on the fate of Si-O bonds which would have been present regardless of the

exact rock composition (see Elements article 10.2113/gselements.12.6.395). I would suggest adding to line 73 the notion that granite is a common rock of the continental crust since the pre-Cambrian, or some other wording that reflects that granite may have been a minor component at the time life arose.

A key aspect of this study is data that suggest that the reactivity of $=\text{Si}^*$ and $=\text{SiO}^*$ has a very different temperature dependence, with $=\text{Si}^*$ being reactive at temperatures well below 100C, while $=\text{SiO}^*$ only become reactive at temperatures around 100C. This mismatch would lead these defects to first form H_2 and then H_2O_2 as they react with water. However, within a closed system the production the two types of defects through breaking of bonds would cancel and eventually all the H_2O_2 formed would be equivalent to H_2 . As suggested in the manuscript, changes in temperature through time and opening the system to fluid flow may create some packages of fluid rich in either H_2 or H_2O_2 . However, this requires a lot of assumptions and unknowns about the early Earth. The formation of H_2O_2 through reaction of water with $=\text{Si-O-O-Si}=\text{}$ that are broken to form $=\text{Si-O}^*$ can provide a sustained production of H_2O_2 .

One issue I do have with the paper is how the authors set out to calculate a global averaged H_2O_2 flux. In my opinion, this is unrealistic. While, it may be useful as a zero-order calculation, it would be more useful to do a calculation of the flux of H_2O_2 into a crack of a given size in the prebiotic crust. For example, one may want to take some data or base some constraints on the studies of the Lost Field site near the Azores. I think that is a far more realistic scenario for Hadean epoch. A warm crust with deep fissures and some connectivity to the overlying ocean. This avoids the reliance on plate tectonics to be active. As the authors write, there is considerable uncertainty on the timing of the onset of plate tectonics and what early stages of plate tectonics may have looked like (see again McLennan and Taylor a perspective on this). Fissures in an pre-tectonic Hadean crust provides a steep gradient hydrothermal temperatures--an environment now seen as a possible location for the emergence of life.

Comment on methods

The methods are very well described, and I compliment the authors on the level of detail provided. One issue I do want to raise is that the analysis of Reactive Oxygen Species is fraught with interferences and complications, particularly if one tries to do this on mineral slurries. The reason is that most of the techniques have been developed for biomedical studies under narrow pH conditions and with a well-defined biofluid composition. There are now several techniques for H_2O_2 detection that have been evaluated for use in mineral slurries. For examples see references immediately below this comment. That still does not guarantee that there may not be any interference. The DMP method used here measures the absorbance of the solution with the reagent at 454nm. One has to establish that the solution without the reagent does not absorb at that wavelength or subtract that signal. Rather than measuring it at one wavelength it is useful to take a scan so interferences are more easily spotted. Also, it is a good research strategy to conduct at least a few experiments in which a second technique is used to corroborate the formation of the ROS. Finally, there is now also a H_2O_2 probe (World Precision) that can provide real-time data. That obviously requires a different setup. But from personal experience, this provides a reliable avenue for obtaining data at lower temperatures. I do not know whether it would

work at the temperatures of interest here, but that may be something to think about for any follow-on work you may contemplate. A student in my lab was able to measure the burst of H₂O₂ forming when minerals were added to an oxygen-containing aqueous solution (i.e., not relevant to this study, but it showed the ability to follow the rapid evolution of H₂O₂ in real-time.)

C.A. Cohn, A. Pak, M.A.A. Schoonen, D.R. Strongin, Quantifying hydrogen peroxide in iron-containing solutions using leuco crystal violet, *Geochemical Transactions* 6(2005) 47–52.

C. Cohn, C. Pedigo, S. Hylton, S. Simon, M. Schoonen, Evaluating the use of 3'-(p-Aminophenyl) fluorescein for determining particulate-induced formation of highly reactive oxygen species, *Geochemical Transactions*, 10, 8 (2009).

C. Cohn, R. Laffers, S. Simon, T. O'Riordan, M. Schoonen, Role of pyrite in formation of hydroxyl radicals in coal: possible implications for human health, *Particle and Fibre Toxicology* 3(2006) 16.

M.A.A. Schoonen, C.A. Cohn, E. Roemer, R. Laffers, S.R. Simon, T. O'Riordan, Mineral-induced formation of reactive oxygen species, *Medical Mineralogy and Geochemistry, Reviews in Mineralogy & Geochemistry* 64, 2006, pp. 179-221

We greatly appreciate the time that reviewers have spent in reviewing this manuscript. Please see below for a detailed point by point response to all comments.

Point by point response to Reviewer #1

1. Surface radicals (e.g., $\equiv\text{Si-O}\bullet$, $\equiv\text{Si}\bullet$, and $\equiv\text{Si-O-O}\bullet$) of silicate minerals are derived from the hemolysis of Si-O-Si bonds. One of the key points for this study is the formation of radicals during ball milling of the rocks. Thus, the authors should provide the evidence for the formation of the corresponding radicals (e.g., $\equiv\text{Si}\bullet$) in the crushed rocks.

We agree that the inclusion of this data would enhance the study. We have now carried out additional crushing experiments, using identical conditions to our original ones, on all 3 rock types and quantified the $\equiv\text{Si}\bullet$ using the DPPH method. These results are given in Supplementary Figure 3, and are incorporated into the main paper as follows (line 83-89):

“Crushing generated similar concentrations of $\text{Si}\bullet$ across all three rock types (ranging from 13.0 to 14.4 $\mu\text{mol g}^{-1}$; Supplementary Figure 3); sufficient to generate the maximum measured H_2 (c. 3 $\mu\text{mol g}^{-1}$; Figure 2) via Eq. 1 and Eq. 2.”

2. In general, the formation of surface radicals in silicate minerals during ball milling strongly depends on Si polymerization degree (the ratio of bridging oxygen per silicon), which commonly decreases in the order of tectosilicate, phyllosilicate, cyclosilicate, inosilicate and nesosilicate. However, this study indicates that peridotite, mainly composed of nesosilicate, has a higher yield of H_2O_2 than granite in which quartz and feldspar (tectosilicate) are main component minerals. The experimental results seem to be unreasonable.

A prior study (Hurowitz, J.A., et al., Production of hydrogen peroxide in Martian and lunar soils. Earth and Planetary Science Letters, 2007. 255(1-2): p. 41-52.) demonstrated the same order of silicate mineral reactivity as we have documented – we have stated this as follows (line 123-129):

“Previous work has suggested a mineral structure control on H_2O_2 generation when silicates are crushed under air, with an inverse relationship between the number of shared corners between silica tetrahedra and H_2O_2 generated^[23]. The mean concentrations of H_2O_2 generated from the three rocks types (peridotite > basalt > granite) in our experiments is broadly consistent with this order, since the dominant minerals in granite such as quartz and feldspars (tectosilicates) have greater numbers of tetrahedra sharing corners than the pyroxenes, amphiboles (inosilicates) and olivine (nesosilicates) present within the basalt and peridotite samples (Supplementary Figure 2).”

We have supported this statement further by conducting XRD analyses (now shown in Supplementary Figure 2) to document the major mineral phases in each of the rock types. We suspect the reason for this order may not only be due to the original number of $\text{SiO}\bullet$ that form, but the reannealing of SiO with adjacent Si to reform Si-O-Si bonds. However, this would take substantial additional work to test and we feel beyond the scope of this manuscript. In any case, as we already state in the discussion we suspect that the dominant source of H_2O_2 in our experiments is not from the $\text{SiO}\bullet$ generated from freshly broken Si-O-Si bonds during cataclasis, but from the pre-existing peroxy defects within the minerals. As such, the concentrations of H_2O_2 generated from each rock may be less influenced by the mineral structure, and more by the original concentration of peroxy defects in the rock.

3. The chemical compositions of the rocks should be provided. Generally, both basalt and peridotite are rich in Fe²⁺ and Mg²⁺, belonging to mafic and ultramafic rocks, respectively. The reaction between basalt/peridotite and water can also produce H₂ (similar to serpentinization). Did the authors consider the contribution of this reaction to the total H₂? More importantly, Fe²⁺ readily reacts with H₂O₂ (i.e., Fenton Reaction), leading to consumption of H₂O₂ and oxidation of Fe²⁺. Thus, the contents of Fe²⁺ in the rocks and solutions have a prominent effect on the quantity of H₂O₂. These factors should be considered in the data analyses and the related discussions.

Dissolved Fe²⁺/Fe³⁺ data was analysed in all of our main experiments at all timepoints. We did not include the data in the original submission as we found no significant correlations with hydrogen generation. We now include all this data in Supplementary Figure 6 and have included the additional statement in the paper as follows (line 89 to 93):

“We do not exclude the possibility that some of the H₂ in experiments was derived from other mineral-water reactions, in particular serpentinization-type reactions of Fe²⁺ with H₂O^[13]. We note however that the maximum aqueous Fe²⁺ concentrations were three orders of magnitude lower than the maximum H₂ concentration, and that there was no significant (p>0.05) correlation between Fe²⁺ and H₂ (R² values ranging from 0.14 to 0.17; Supplementary Figure 6).

We also now include the mineralogical composition of all samples in Supplementary Figure 2.

For the potential for Fenton reactions, we have now included in the discussion new data (Supplementary Figures 14 and 15) and the following adapted paragraph (line 218 to 228):

“We note as well that our measured H₂O₂ concentrations may underestimate total production due to subsequent decomposition,^[24] particularly if the decomposition is accelerated by higher temperatures and/or the presence of Fe species via Fenton reactions (Araujo et al 2011). The presence of low concentrations of detectable dissolved iron species (up to nearly 0.037 μmol g⁻¹ (18.5 μM) in peridotite experiments) suggests the potential for Fenton reactions (Eq. 14, Eq. 15) to have accelerated the removal of H₂O₂ (Eq. 14, Eq. 15)^[27], although there were no clear trends with time or temperature in the Fe data (Supplementary Figure 14), nor any significant relationship between Fe²⁺ and H₂O₂ concentrations (Supplementary Figure 15).

Point by point response to Reviewer #2

1. p. 6, l. 164: The wording “[...] was warm to the touch [...]” seems somewhat inaccurate. It would be better to give at least an estimate of the temperature to which the ball mill heated.

We agree. We have therefore now measured the temperature of the ball mill after replicating the crushing procedures used in the experiments, and added the following sentence (line 179 to 181): “After our planetary ball milling (30 min at 500 rpm) the bulk temperatures of the crushed rocks increased by 10.7 ± 0.6 ° C (Methods).”

2. Discussion on p. 9: The H₂O₂ produced by cataclasis could also be involved in the prebiotic synthesis of e.g. amino acids. Experiments on the synthesis of amino acids under hydrothermal conditions have also shown that

the addition of O₂/H₂O₂ can also increase the rates of abiotic synthesis of selected amino acids and allow the synthesis of others, especially amino acids with additional hydroxyl groups, such as Ser, Lys and Thr (Marshall 1994, *Geochem. Cosmochim. Acta* 58: 2099). Therefore, a role of local microquantities of O₂/H₂O₂ in the processes leading to the prebiotic synthesis of important biopolymers cannot also be completely excluded.

Thank you. We have now added a sentence at the end of the discussion to include the synthesis of amino acids (line 320-321).

3. A minor methodological note concerns the representation of rock crushing in a ball mill in revolutions per minute ("rpm") and not in "g" (g-force, the relative centrifugal force [RCF]). In centrifuges, "g" is a more universal unit, as "rpm" depends on the radius of the rotor used. I am not sure, but perhaps for ball mills only "rpm" is used as a unit.

For completeness, we have now also added the g value (line 339).

4. Add a legend to Supplementary Figure 2.

We have now added a legend.

Response to Reviewer #3

I would suggest adding to line 73 the notion that granite is a common rock of the continental crust since the pre-Cambrian, or some other wording that reflects that granite may have been a minor component at the time life arose.

We have changed the wording on granite as suggested (line 71 to 73):

“To test this further, we carried out additional experiments crushing not only granite (a common rock in the continental crust since the Precambrian) but also basalt and peridotite (representing oceanic crust).”

One issue I do have with the paper is how the authors set out to calculate a global averaged H₂O₂ flux. In my opinion, this is unrealistic. While, it may be useful as a zero-order calculation, it would be more useful to do a calculation of the flux of H₂O₂ into a crack of a given size in the prebiotic crust. For example, one may want to take some data or base some constraints on the studies of the Lost Field site near the Azores. I think that is a far more realistic scenario for Hadean epoch. A warm crust with deep fissures and some connectivity to the overlying ocean. This avoids the reliance on plate tectonics to be active. As the authors write, there is considerable uncertainty on the timing of the onset of plate tectonics and what early stages of plate tectonics may have looked like (see again McLennan and Taylor a perspective on this). Fissures in an pre-tectonic Hadean crust provides a steep gradient hydrothermal temperatures--an environment now seen as a possible location for the emergence of life.

We have included the global H₂O₂ flux estimates in order to put them in context of other global fluxes, including those of atmospheric UV generated H₂O₂ fluxes. To clarify the discussion on the early Earth, we have now more clearly separated our estimates of modern day H₂O₂ fluxes (where there is sufficient data available to come up with at least some very broad estimates based on the modern plate tectonic regime, including estimates of H₂O₂ concentrations in fractures based upon prior calculations of Hirose et al. 2011, and where we now explicitly mention the Lost City hydrothermal field), and a later discussion on early prebiotic Earth. For the latter, we feel

that there are simply too many unknowns/uncertainties to attempt a realistic estimate of Hadean/early earth fluxes wither globally or within fractures. Rather, we now state (lines 284-298).

“To put the modern day estimates of H₂O₂ generation in context of the early Earth, the estimated maximum present day surface normalised fluxes from catalclasis (1.1×10^9 molecules H₂O₂ cm⁻² s⁻¹) and peroxy linkages (6.4×10^8 molecules H₂O₂ s⁻¹ cm⁻²) are 1000 × and 640 × greater than that estimated to have been produced in the pre-photosynthetic Archean atmosphere by UV photochemical reactions (10^6 molecules H₂O₂ cm⁻² s⁻¹)^[3]. Clearly, however, the tectonic regime of the early Earth would have been very different to that of the present day. The date at which global plate tectonics similar to that of the present day first started is still an area of controversial active research, with estimates between 3.8 to 0.7 Ga depending on different lines of evidence^[36]. Before global plate tectonics commenced, localised rates of subsurface stressing and fracturing, and hence potential H₂O₂ generation, would still have been driven via more localised vertical tectonics^[37], magma intrusion^[38], and meteorite strikes^[23]. While the composition of Hadean and early Archean crust would also have been very different (including rock types such as ultramafic komatiites that are not formed in the present day^[39]), the experimental generation of H₂O₂ generated from modern rock types spanning felsic to ultramafic (Figure 2) suggest that oxidant production from catalclasis and peroxy linkages should also be relevant to other silicate rocks.”

The key point is that our proposed mechanisms of subsurface H₂O₂ generation would have been present on the early Earth. As to how quantitatively large they were, we will have to leave for future work.

The methods are very well described, and I compliment the authors on the level of detail provided. One issue I do want to raise is that the analysis of Reactive Oxygen Species is fraught with interferences and complications, particularly if one tries to do this on mineral slurries. The reason is that most of the techniques have been developed for biomedical studies under narrow pH conditions and with a well-defined biofluid composition. There are now several techniques for H₂O₂ detection that have been evaluated for use in mineral slurries. For examples see references immediately below this comment. That still does not guarantee that there may not be any interference. The DMP method used here measures the absorbance of the solution with the reagent at 454nm. One has to establish that the solution without the reagent does not absorb at that wavelength or subtract that signal. Rather than measuring it at one wavelength it is useful to take a scan so interferences are more easily spotted.

We have now included full wavelength scans (Supplementary Figure 7) showing a very close match between an H₂O₂ standard and a representative high H₂O₂ sample (crushed peridotite, 104°C, 1 week). This gives us confidence that we are measuring H₂O₂.

Also, it is a good research strategy to conduct at least a few experiments in which a second technique is used to corroborate the formation of the ROS. Finally, there is now also a H₂O₂ probe (World Precision) that can provide real-time data. That obviously requires a different setup. But from personal experience, this provides a reliable avenue for obtaining data at lower temperatures. I do not know whether it would work at the temperatures of interest here, but that may be something to think about for any follow-on work you may contemplate. A student in my lab was able to measure the burst of H₂O₂ forming when minerals were added to an oxygen-containing aqueous solution (i.e., not relevant to this study, but it showed the ability to follow the rapid evolution of H₂O₂ in real-time.)

We thank the reviewer for their expertise and advice on this. Going forward, I am happy to say that we have just had a three year grant funded that has, amongst other eqpt included this exact apparatus (World Precision H₂O₂ microsensor) for follow up work. However, this has yet to be ordered, and is we feel beyond the scope of the current manuscript as we would have to repeat the experiments (which were primarily generated over the course of a one year Masters level project) again. While we agree it would be ideal to run more than one

method to quantify H_2O_2 , we feel that the wavelength scans we have now included in response to the reviewer's justified comments (Supplementary Figure 7) do give further confidence that we are measuring the same complex as in the standards. The dichotomy between the presence of H_2 and H_2O_2 (e.g. see Supplementary Figure 8a, b, c) is also consistent with the presence of H_2O_2 , and not simply the presence of a universal interfering chemical species within the experiments which might not be expected to generate such strong relationships.

REVIEWERS' COMMENTS

Reviewer #1 (Remarks to the Author):

Overall, the authors have strengthened the manuscript and addressed the reviewer comments clearly.

However, there is still one important issue that needs to be addressed clearly to improve the manuscript. In the balling mill experiments, the agate mortar was used for rock crushing. It is noteworthy that the Mohs' hardness of agate (6.5-7) is similar to those of peridotite (6.5-7), basalt (5-7) and granite (6-7). This means that agate powder (it is composed of SiO_2) will be produced during balling mill, and the resulting agate powder will also contribute the generation of H_2O_2 and H_2 . The authors should check the experiments carefully and exclude the contribution from the agate powder.

Other comments:

1. As there is considerable uncertainty on the timing of the onset of plate tectonics, and this paper focuses on the effect of temperature on the oxidant-producing processes, it seems that a title of "Geothermally driven oxidant production in the hot biosphere" is more suitable.
2. Please check the equation numbers carefully, e.g., line 245, line 341.

Reviewer #3 (Remarks to the Author):

I have now reviewed the revised manuscript and carefully read the cover letter that spells out how the authors have addressed all reviewer comments.

The authors have carefully and thoughtfully responded to comments. The article represents an important new perspective on ROS formation in the absence of O_2 . The experiments are carefully designed and executed. While the title focuses on a driving role for tectonic, the formation mechanism is not dependent on it, but the flux may be enhanced by it. I think this paper will be of broad interest and spur other work.

One minor thing. I think that Tectonically driven should be hyphenated that is Tectonically-driven.

Re: Manuscript NCOMMS-22-02775-T

We greatly appreciate the time that reviewers have spent in reviewing this manuscript a second time. Please see below for a detailed point by point response to all comments.

Response to Reviewer #1

“However, there is still one important issue that needs to be addressed clearly to improve the manuscript. In the balling mill experiments, the agate mortar was used for rock crushing. It is noteworthy that the Mohs’ hardness of agate (6.5-7) is similar to those of peridotite (6.5-7), basalt (5-7) and granite (6-7). This means that agate powder (it is composed of SiO₂) will be produced during balling mill, and the resulting agate powder will also contribute the generation of H₂O₂ and H₂. The authors should check the experiments carefully and exclude the contribution from the agate powder.”

To quantitatively address this point, we have carried some further milling experiments where we have measured the mass loss of the agate ball mill after milling under the same conditions as our original experiments (45 g of rock, 500 rpm for 30 min). Results are now stated in lines 88/89 “...with estimates of contamination from the agate ball mill and agate grinding balls to the rock powders $\leq 0.2\%$ (Supplementary Table 5).” Full data is shown in Supplementary Table 5, and full methods are given in Methods. We note that the very low potential agate contamination is most likely due to the high amount of material (45 g) used in our milling. The instructions for the ball state that at least 10 g should be used to limit excessive abrasion. The higher amounts we used (effectively the maximum that can be used) would have quickly generated powder over the agate grinding balls and shielded the agate from direct abrasion.

“As there is considerable uncertainty on the timing of the onset of plate tectonics, and this paper focuses on the effect of temperature on the oxidant-producing processes, it seems that a title of “Geothermally driven oxidant production in the hot biosphere” is more suitable.”

Re: the title, we have added in a hyphen (Tectonically-driven). We would contend that the title is accurate as is, for the reason that ‘tectonics’ is not a synonym for ‘plate tectonics’. Plate tectonics infers the horizontal global movement of crustal plates. Tectonics refers to any structural movement, which could be induced by e.g. sediment deposition, hot spot volcanism, or meteorite impacts. Tectonics would certainly have been present prior to the advent of plate tectonics on the early Earth for these reasons. We feel that the alternative suggested title ‘Geothermally-driven oxidant production in the hot biosphere’ does not give a full picture of our study. Rather, it is the combination of tectonically induced stress and hot temperatures which is required to release oxidised defects from rocks and react them with water to produce hydrogen peroxide – not hot temperatures alone.

“Please check the equation numbers carefully, e.g., line 245, line 341.”

Thank you – yes you are correct, two Equations (now correctly labelled 16 and 17) were incorrectly labelled. This has now been corrected.

Response to Reviewer #3

“One minor thing. I think that Tectonically driven should be hyphenated that is Tectonically-driven.”

Thank you – we agree. We have now hyphenated the title as suggested.